DATA RELEASE

# Spatial patterns associated with the distribution of immature stages of *Aedes aegypti* in three dengue high-risk municipalities of Southwestern Colombia

Cristina Sánchez Gutierrez[1], Erika Santamaría[1], Carlos Andrés Morales[2], María Camila Lesmes[1,3], Horacio Cadena[4], Alvaro Avila-Diaz[3,5], Patricia Fuya[1] and Catalina Marceló-Díaz[1,*]

1 Grupo de Entomología, Instituto Nacional de Salud, Bogotá, Colombia
2 Secretaría de Salud Departamental del Cauca, Popayán, Colombia
3 Universidad de Ciencias Aplicadas y Ambientales, Bogotá, Colombia
4 Programa de Estudio y Control de Enfermedades Tropicales PECET, Medellín, Colombia
5 Research Group "Interactions Climate-Ecosystems (ICE)", Earth System Science Program, Faculty of Natural Sciences, Universidad del Rosario, Bogotá, Colombia

**Submitted:** 02 May 2023

* Corresponding author. E-mail: cmarcelo@ins.gov.co; catalina.marcelo@udea.edu.co

Preprint submitted at https://doi.org/10.1590/SciELOPreprints.6611

Included in the series: *Vectors of human disease* (https://doi.org/10.46471/GIGABYTE_SERIES_0002)

## ABSTRACT

*Aedes aegypti* mosquitoes are the main vector of human arbovirosis in tropical and subtropical areas. Their adaptation to urban and rural environments generates infestations inside households. Therefore, entomological surveillance associated with spatio-temporal analysis is an innovative approach for vector control and dengue management. Here, our main aim was to inspect immature pupal stages in households belonging to municipalities at high risk of dengue in Cauca, Colombia, by implementing entomological indices and relating how they influence adult mosquitos' density. We provide novel data for the geographical distribution of 3,806 immature pupal stages of *Ae. aegypti*. We also report entomological indices and spatial characterization. Our results suggest that, for *Ae. aegypti* species, pupal productivity generates high densities of adult mosquitos in neighbouring households, evidencing seasonal behaviour. Our dataset is essential as it provides an innovative strategy for mitigating vector-borne diseases using vector spatial patterns. It also delineates the association between these vector spatial patterns, entomological indicators, and breeding sites in high-risk neighbourhoods.

**Subjects** Human and Biomedical Sciences, Global Health, Biomedical Science

# DATA DESCRIPTION

## Introduction

The *Aedes aegypti* mosquito is the main vector of human arbovirosis, encompassing diseases such as yellow fever, dengue, chikungunya, and Zika, which are prevalent in tropical and subtropical areas [1]. Its adaptation to local anthropic conditions, urban environments, together with the optimal temperature, precipitation, and relative humidity, among other environmental and socioeconomic factors, has given this mosquito the ability to establish reproductive populations spanning both urban and rural domains [2–4]. In Colombia, *Ae. aegypti* is the main vector of the dengue virus and is reported at altitudes

between sea level and 2,302 m. Therefore, the significant presence and abundance of this vector threaten public health within the country [5, 6].

Direct proximity to humans has allowed the hematophagous female to feed on human or domestic animal blood, causing infestations in different types of containers in and around households. In this context, the presence of artificial and natural containers, such as tanks, drums, bottles, tires, aquatic plants (bromeliads and water retained in plant axils), stagnant water, and pet drinking troughs, serve as potential breeding grounds for the immature stages of *Ae. aegypti* [7, 8]. The surveillance of these types of containers helps reduce the proliferation of pupae and, therefore, mosquito densities. Hence, mosquito sampling is an indispensable tool for designing approaches of entomological surveillance of the vector specific to each situation [9].

Following the methodologies established for Colombia [7, 10], the inspection of potential breeding sites is determined with entomological indicators that make it possible to identify the index of pupae per person, pupal productivity, and the presence of immature stages in dwellings [7, 10]. However, this is not done in all regions. The importance of using this type of entomological index lies in providing a representative approximation of the local adult populations of mosquitoes while requiring a relatively small sample size for household inspection [9, 11]. Furthermore, the index of pupae per person makes it possible to calculate a minimum level of *Ae. aegypti* pupae infestation that significantly enhances the risk of dengue virus (DENV) transmission. Specifically, if the index of pupae per person ranges from 0.5 to 1.5 at 28 °C with 0% to 67% seroprevalence [12], making it possible to identify areas at higher risk.

Surveillance of *Ae. aegypti* mosquitoes has mainly relied on qualitative larval indices, bearing little relation to the number of female mosquitoes. Moreover, *Ae. aegypti* populations have been reported to be spatially heterogeneous in some areas [11]. Thus, using spatio-temporal analyses to predict *Ae. aegypti* hot spots, i.e., groups of dwellings with higher pupal productivity, high vector density, and potential DENV transmission, is an innovative strategy that could help the prevention, vector control, and integrated management of dengue in high-risk municipalities. This is because this method allows detecting patterns of geographical distribution of the vector [11, 13]. Consequently, it is fundamental to embrace this kind of infestation indices because immature stages, specifically the pupae, represent a precursor stage to adult mosquitos with very low mortality. This increases the correlation between *Ae. aegypti* density and dengue virus [7, 14].

Thus, in this study, entomological surveillance was compared using quantitative indices of pupae and adults in three municipalities at high dengue risk: Patía (El Bordo), Miranda, and Piamonte in the Cauca department. Specifically, our study aimed at examining the influence of pupal productivity, among other entomological indices, on the density of adult mosquitoes, and their spatial and temporal patterns. Spatial regression and bivariate spatial autocorrelation analyses were carried out, allowing to model and explore spatial relationships, while capturing the relationship between variables even when they did not overlap within the same space.

## CONTEXT

A total of 160 biological records (3,806 specimens) of the immature stage (pupae) of *Ae. aegypti* were collected from dwellings in the municipalities of Patía (El Bordo), Miranda,



and Piamonte, within the department of Cauca, in the southwest of Colombia. These municipalities are part of an ongoing research project titled "Spatial stratification of dengue based on the identification of risk factors: a pilot trial in the department of Cauca" headed by the Entomology Group of the National Institute of Health, the Secretary of Health of Cauca, and the University of Applied and Environmental Sciences (UDCA). Among other objectives, this project seeks to evaluate the robustness of the link between entomological variables, which explains the spatial pattern observed in the distribution of dengue fever.

The data were collected between 2021 and 2022 by a multidisciplinary team of environmental health technicians, geographic and environmental engineers, and professionals with extensive experience in medical entomology. In this dataset, we distinguish three sampling periods for each municipality. For Patía: March and October 2021, and March and April 2022; for Miranda: April and November 2021, and May 2022; and for Piamonte, July 2021, and February and July 2022. A fourth sampling period was performed in August and October 2022 in the municipalities of Patía and Miranda, respectively, to increase the sample size to carry out the serotyping of the specimens.

The immature stages (pupae) were collected from the inspection of containers smaller or larger than 20 l, including tanks, drums, and tires, among others, from inside the houses of three municipalities. The municipalities were selected as study areas based on their endemic-epidemic behaviours for DENV transmission, as they are characterized by focal endemics, heterogeneous transmission scenarios, and temporal and cyclical patterns of at-risk populations.

The resulting pupae dataset is in Darwin Core file format, with 73 terms available. We included all mandatory fields, which were submitted to the Integrated Publishing Toolkit (IPT) for review by SiB-Colombia. Metadata fields are also available from the IPT website [15]. A total of 3,480 adult individuals of the family Culicidae are included in this dataset. Of these, about 69% of the specimens were previously reported in 2021 [16], while 1,078 are the new records collected in 2022.

These biological registers are fundamental for the scientific community as they provide the geographical coordinates and entomological indices necessary for the public health surveillance performed by health entities. In particular, these measures are essential for preventing and controlling the etiological agent from houses and neighbourhoods with high infestation rates.

## METHODS

### Sampling

The specimens of the immature stages of the *Ae. aegypti* pupae belonged to dwellings located in the municipalities of Patía (El Bordo), Miranda, and Piamonte, within the department of Cauca, Colombia (Figure 1). These municipalities were selected because they were considered endemic-epidemic with a high dengue transmission risk after conducting a spatiotemporal analysis [16]. The sample size was delimited spatial scale of blocks by Kernel density analysis, georeferencing the dengue cases reported from 2015 to 2019 in the urban areas of each municipality. Additionally, the sample size was calculated from the estimated prevalence of dengue (10.5%) in the municipality [17] with a confidence level of 99% calculated by the Epi Info™ software, using the estimated population size and the clusters obtained in the Kernel analysis.

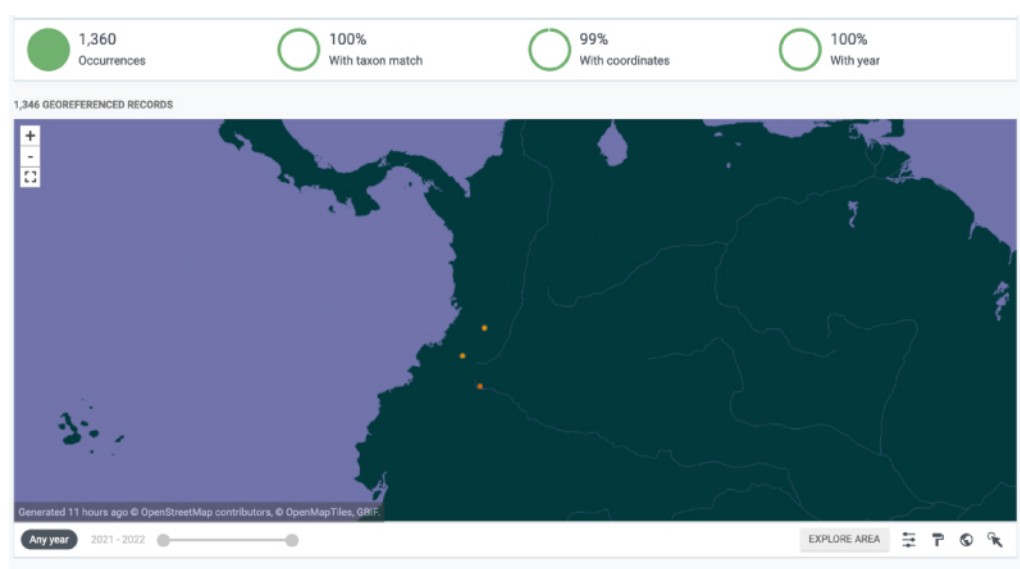

**Figure 1.** Interactive map of the georeferenced occurrences hosted by GBIF [18]. https://www.gbif.org/dataset/a51b1fb9-6e67-42b9-879b-5f21e4d85642

In total, 1,919 dwellings were visited in 26 neighbourhoods during March, April, July, and October 2021, and February, March, May, July, August, and October 2022. In the municipality of Patía, 11 neighbourhoods were visited (*n* = 580 households), in Miranda 12 neighbourhoods (*n* = 854 households), and in Piamonte four neighbourhoods (*n* = 485 households).

## Species collection

A survey of the dwellings was carried out during the day between 8:00 a.m. and 5:00 p.m. Then, the parameters stipulated in the guide Management for the entomological surveillance and control of dengue transmission were followed [7, 10]. Artificial water containers found in the households were selected. All breeding sites were inspected regardless of their size, i.e., containers smaller than 20 l, such as bottles, vases, cans, tires, and small plastic containers. The pupae were collected with a Pasteur pipette and then counted. For containers larger than 20 l, the number of pupae collected with a sweep net was counted, and the volume of the tanks for water storage (TWS) and the existing water level were evaluated to calculate the calibration factor [7] and obtain the estimated number of pupae or the pupal productivity.

Entomological inspections for adult mosquitoes were conducted during specific months in 2021 (February, March, April, July, October, and November) and 2022 (February, March, May, and July). These inspections were performed between 8:00 a.m. and 5:00 p.m., averaging 10 min per house. In each dwelling, a search for adult mosquitoes was conducted in the living room, dining room, bathrooms, kitchens, laundry yard, and other areas; the mosquitoes were captured using a Prokopack aspirator. Using these procedures, during the examination, special attention was directed toward exploring shaded areas and locations near water containers.

**Table 1.** Indicators for *Ae. aegypti* entomological surveillance [7, 9, 10].

| Index | Calculation | Interpretation |
|---|---|---|
| Pupal productivity | (Number of pupae in the container type * calibration factor) | Calculate the estimated number of pupae per container. For deposits smaller than 20 l, only the number of pupae is counted. For deposits larger than 20 l, multiply the number of pupae by calibration factor (c.f.) provided by water level. **Water levels:** <1/3 c.f. not applicable; 1/3 c.f.: 2.6; 2/3 c.f.: 3.0; 3/3 c.f.: 3.5. |
| Female pupae productivity | (Number of pupae assuming a 1:1 sex ratio at emergence) | Estimate the number of emerging adult mosquito females produced per container. |
| Pupae-per-person index | (Pupal productivity/total population of the screened houses) | Generate an estimate of the number of pupae per person in the screened household. |
| Breteau Index | (Number of containers with any pupae *100/Number of screened houses) | Defined as total number of positive containers per 100 households inspected. Determined by a ratio of positive containers to screened households. |

## Species classification and spatial characterization

After the collection of the immature stages (pupae) and adults, the species were identified and taxonomically classified by entomological experts supported by the taxonomic keys developed by Forattini [19] (1995) and Harrison *et al.* [20] (2016) to differentiate them from the immature stages and adults belonging to other species. Continuing with the field protocol, 3,806 pupae of *Ae. aegypti* were identified, of which 395 were preserved in 0.2 ml vials with RNAlater© for subsequent processing with molecular biology techniques at the Entomology group and the Genomics of Emerging Microorganisms group of the National Institute of Health (Bogotá, Colombia) [21].

The entomological information collected was recorded using the ArcGIS® Survey 123 application [22], which provided the geographic location of each specimen collected. A code was associated to each vial with the socio-demographic information of the survey. The total number of pupae recorded in each survey was used to determine the entomological indices of pupal productivity, female productivity, pupal index per person, Breteau's pupal index [7, 10], and *Aedes* sp. pupae sex ratio F:M (Table 1).

The entomological information, together with the geographical block information of the municipalities, was used to perform scatterplot analyses ($R^2$, slope *b*, *p*-value), as well as scatterplot matrices to evaluate the relationships between entomological variables: number of adult *Ae. aegypti* mosquitoes [18], total number of pupae, pupal productivity, female productivity, and number of pupae per person vs. the frequency of tanks for water storage, miscellaneous containers smaller than 20 l, and drums.

A global bivariate Moran Index analysis ($p \leq 0.05$) was performed to detect the distribution of variables related to spatial clustering. The Moran Index ranges from −1 to 1, where −1 indicates dispersed clustering patterns, 0 indicates randomness, and 1 suggests perfect association. Next, a Local Moran Index analysis distributed the significant ($p = 0.05$) clusters of dwelling blocks into four types of local spatial association. (I) high(*x*)-high(*y*) indicates areas with high values of the variable (*x*) surrounded by values above the mean of the variable under analysis (*y*); (II) low(*x*)-high(*y*) indicates areas with low values of the variable (*x*) surrounded by neighbouring areas with values above the mean of the variable (*y*); (III) low(*x*)-low(*y*) indicates areas with low values of the variable (*x*) surrounded by areas with values below the mean of the variable (*y*); and (IV) high(*x*)-low(*y*) indicates areas with high values of the variable (*x*) surrounded by areas below the mean of the variable (*y*) [13].

**Table 2.** Descriptive entomological measures by sampling locality. The total of positive screened houses for each mosquito species is shown as a total and as a percentage for each municipality [24, 25].

| Entomological measure | Municipality | | | Total |
|---|---|---|---|---|
| | **Patía** | **Miranda** | **Piamonte** | |
| Number of houses screened | 580 | 854 | 485 | 1919 |
| Number of habitants in screened houses | 2098 | 3735 | 1880 | 7713 |
| # of positive houses with *Aedes* sp. pupae (%) | 67 (11.55%) | 58 (6.79%) | 35 (7.22%) | 121 |
| Number of containers screened | 1142 | 1067 | 984 | 3193 |
| # of positive containers with *Aedes* sp. Pupae (%) | 50 (4.38%) | 43 (4.03%) | 36 (3.66%) | 129 |
| Total #of *Aedes* sp. pupae | 425 | 530 | 389 | 1344 |
| *Aedes* sp. pupae productivity | 1493 | 1173 | 1140 | 3806 |
| *Aedes* sp. pupae sex ratio F:M [21, 22] | 1:1 | 1:1 | 1:1 | 1:1 |
| Number of pupae per person | 0.71 | 0.31 | 0.61 | 1.63 |
| Breteau index | 8.62 | 5.04 | 7.42 | 21.08 |

The above analyses were performed for two sampling periods in each municipality. For the municipality of Patía (El Bordo), samples were collected in March and October 2021, for Miranda samples were collected in April and November 2021, and for Piamonte samples were collected in July 2021 and February 2022, since the rainfall pattern was seasonal between these months. Finally, the analyses were performed using the GeoDa v1.20 program [23] and the maps were visualized using the ArcGIS® 10.8 software (RRID:SCR_011081).

## DATA VALIDATION AND QUALITY CONTROL

A total of 3,480 adult specimens were identified across 1,200 households, comprising 1,459 females and 2,021 males. In 2021, 2,402 records were documented, while 1,078 new records were added in 2022. In terms of municipal distribution, Miranda reported 500 individuals (14.36%), Patía 1,305 (37.5%), and Piamonte had the highest count with 1,675 (48.13%) adult individuals. Notably, 71 individuals were found within a Patía household, and an additional 125 individuals were identified within a different household in Piamonte.

Moreover, a total of 3,806 immature specimens (pupae) of *Ae. aegypti* were identified, distributed among 160 records (positive dwellings out of the total inspected). For the municipality of Patía (El Bordo), 1,493 individuals were found in 67 positive dwellings, for Miranda 1,173 individuals were in 58 dwellings, and for Piamonte 1,140 individuals were found in 35 dwellings. The neighbourhoods with the highest number of specimens were Villa Los Prados (*n* = 519) and La Paz (*n* = 359) in Piamonte, followed by the San Antonio neighbourhood (*n* = 391) in Miranda, and the Libertador (*n* = 441) and Olaya Herrera (*n* = 438) neighbourhoods in Patía. It should be noted that 70 of the inspected dwellings were found with more than 15 pupae inside the house. The largest number of detected specimens belonged to a house in the municipality of Piamonte, with 280 pupae (Table 2).

For the wet season of the sampling, our spatial regression analysis found a positive correlation for the municipalities. Specifically, the determination coefficient between the captures of pupae and adults was higher than 50% in the municipalities of Patía (*R* = 0.7292; $R^2$ = 0.53) and Miranda (*R* = 0.7486; $R^2$ = 0.56) and lower in Piamonte (*R* = 0.4252; $R^2$ = 0.18), highlighting the relevance of the adult index for entomological surveillance. In addition, a spatial autocorrelation was observed between the presence of pupa-positive houses and a higher density of adults in neighbouring blocks.

For the Patía municipality, the variable productivity of female pupae explained the number of adult mosquitoes with a model adjustment of 53.2% in the period of higher

**Table 3.** Results of the regression analysis and the bivariate spatial autocorrelation for entomological variables of *Ae. aegypti* in Patía, Miranda, and Piamonte, according to the seasonality of sampling.

| Municipality | Seasonality | Variables (*X*, *Y*) | Parameters | | | Moran I (MI) |
|---|---|---|---|---|---|---|
| | | | $R^2$ | Slope *b* | *p* value | |
| Patía (El Bordo) | Wet season higher precipitation March 2021 | % TWS - PPP | 0.232 | 0.082 | 0.000* | **0.264** |
| | | Pupal productivity ♀ - PPP | 0.327 | 0.313 | 0.000* | **0.325** |
| | | % TWS - No. adult mosquitoes | 0.126 | 0.053 | 0.006* | **0.132** |
| | | % TWS - Pupal productivity ♀ | 0.374 | 0.191 | 0.000* | **0.109** |
| | | Pupal productivity ♀ - No. adult mosquitoes | 0.532 | 0.530 | 0.000* | **0.108** |
| | Wet season lower precipitation October 2021 | % TWS - PPP | 0.560 | 0.106 | 0.000* | **−0.048** |
| | | Pupal productivity ♀ - PPP | 0.680 | 0.262 | 0.000* | **0.034** |
| | | % Flower vases - PPP | 0.262 | 0.587 | 0.000* | **0.033** |
| | | % Flower vases - Pupal productivity ♀ | 0.150 | 0.701 | 0.004* | **0.115** |
| | | Pupal productivity ♀ - No. adult mosquitoes | 0.084 | 0.145 | 0.035* | **0.196** |
| Miranda | Wet season lower precipitation April 2021 | Pupal productivity ♀ - No. adult mosquitoes | 0.56 | 0.114 | 0.000* | **0.086** |
| | | % TWS - No.adult mosquitoes | 0.116 | 0.07 | 0.001* | **0.051** |
| | | Pupal productivity- No. adult mosquitoes (♀) | 0.437 | 0.028 | 0.000* | **0.043** |
| | Wet season higher precipitation November 2021 | Pupal productivity ♀ - No. adult mosquitoes | 0.043 | 0.366 | 0.064 | **−0.035** |
| | | % TWS- number of adult mosquitoes | 0.000 | -0.003 | 0.965 | **−0.029** |
| | | Pupal productivity - No. adult mosquitoes (♀) | 0.045 | 0.145 | 0.058* | **−0.047** |
| Piamonte | Wet season higher precipitation July 2021 | Pupal productivity ♀ - No. of adult mosquitoes | 0.181 | 0.037 | 0.004* | **0.058** |
| | | % miscellaneous containers < 20L - PPP | 0.001 | -0.063 | 0.809 | **0.054** |
| | | % drums-Pupal productivity ♀ | 0.054 | 1.026 | 0.123 | **0.043** |
| | Dry season lower precipitation February 2022 | Pupal productivity ♀ - No. adult mosquitoes | 0.197 | 0.085 | 0.005* | **0.044** |
| | | % miscellaneous containers 20L - PPP | 0.006 | 0.004 | 0.643 | **0.124** |
| | | % drums-Pupal productivity ♀ | 0.107 | 0.247 | 0.042* | **0.112** |
| | | % buckets-Pupal productivity ♀ | 0.377 | 1.357 | 0.000* | **−0.026** |

PPP: pupae per person, TWS: tanks for water storage, * *p* ≤ 0.05.

rainfall and with an adjustment of 8.4% in the period of lower rainfall, presenting in both cases positive autocorrelation between the variables (Table 3). This allowed us to locate the clusters in which pupal productivity generated a high mosquito density at the block level (Figure 2).

Variables such as the frequency of positive vases when rainfall reached only 1 mm (October 2021) explained the female pupal productivity by only 15%. However, in the period of higher precipitation (211 mm, March 2021), the tank container explained 37% of the female pupal productivity and, subsequently, the density of adult mosquitoes. This comparison is interesting, considering that the climate of the Patía municipality is bimodal, with peaks in April and November [26]. Our local analysis (LISA) observed heterogeneous high-high and low-low spatial clusterings for each season, mainly in the Olaya Herrera neighborhood (March and October 2021).

The model for the municipality of Miranda showed an $R^2$ model fit of 56% and a positive spatial autocorrelation in the season of lower rainfall, although with a high percentage of relative humidity, among the variables of female pupal productivity, explaining the number of adult mosquitoes. Positive autocorrelation was also found for the variables frequency-of-positive-low-TWS and pupae-productivity. However, for the period of higher precipitation and lower relative humidity, a negative spatial autocorrelation was found for the variables evaluated, i.e., with clustering patterns close to randomness (Table 3).

For Piamonte, during the wet season, the female-pupal-productivity variable explained the model fit ($R^2$) by 18%, with a positive bivariate spatial autocorrelation (Moran Index (MI) = 0.058). Miscellaneous containers smaller than 20 l, drums, and TWS presented non-significant positive autocorrelation of entomological indicators and number of adults. This latter finding is in contrast with the correlation between the percentage of positive buckets and pupae productivity ($R^2$ = 37.7%) in the dry season of sampling.



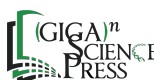

**Figure 2.** Bivariate local Moran index for the variables pupal productivity (*x*) and number of adult mosquitoes (*y*) in the municipalities of Patía, Miranda, and Piamonte (2021–2022).

These findings suggest that for the species *Ae. aegypti*, pupal productivity generates high densities of adults in neighbouring houses, allowing us to identify with the local indicator of spatial association (LISA) the blocks of neighbourhoods where this trend occurs, evidencing seasonal behaviour. While the spatial correlation linked the capture of pupae and adults in the same geographical space, the bivariate autocorrelation (MI) related these same variables, without necessarily coinciding in the same dwelling. Likewise, between municipalities, a greater positive spatial autocorrelation between pupae and adults was observed in Patía, especially when precipitation decreased before the onset of the rainy season. Our results are similar to other spatiotemporal studies of mosquito density [13, 27], showing heterogeneous patterns of occurrence for each territory, with seasonal behaviours registering higher infestation rates in seasons of higher rainfall.

## RE-USE POTENTIAL

Our database and vector distribution map provide important resources for understanding the spatial patterns of the vector and its relationship with entomological indicators and breeding sites, which could increase dengue virus transmission in the municipalities.

To improve the accessibility and usability of these data, they have been included in the GBIF. These data will be useful for making representative approximations of mosquito densities, mapping areas with high increases in pupal productivity, and linking other environmental, entomological, or socio-demographic determinants, providing essential information to generate innovative strategies for prevention, vector control, and management of dengue. We suggest others make their data available as well.

## DATA AVAILABILITY

The datasets on which this article is based are available in the GBIF repository [18] as well as in GigaDB [28].

## EDITOR'S NOTE

This paper is part of a series of Data Release articles working with GBIF and supported by TDR, the Special Programme for Research and Training in Tropical Diseases, hosted at the World Health Organization [29].

## ABBREVIATIONS

DENV, dengue virus; IPT, Integrated Publishing Toolkit; LISA, local indicator of spatial association; MI, Moran Index; TWS, tanks for water storage; UDCA, University of Applied and Environmental Sciences.

## DECLARATIONS

### Ethics approval and consent to participate

This work was ethically approved by the Comité de Etica y Metodologías de la Investigación (CEMIN) of the National Institute of Health in Bogotá, Colombia (Research Project CEMIN 13-2019).

### Consent for publication

Not applicable.

## Competing Interests

The authors declare that they have no competing interests.

## Authors' contributions

CS: data curation, writing (original draft, editing), data analysis, software, investigation; ES: investigation, supervision, project administration, field data, writing (review and editing); CAM: conceptualization, project administration, field data, writing (review and editing); MCL: field data, data curation, visualization; HC: writing (review and editing); AAD: writing (review and editing); PF: writing (review and edit, field data; CMD: conceptualization, data curation, software, visualization, writing (original draft, editing), methodology, project administration, funding acquisition, investigation. All authors made comments on the manuscript.

## Funding

This work was supported by the National Institute of Health, Bogotá, Colombia (research project CEMIN 13-2019), the Secretaria de Salud Departamental del Cauca, the Universidad de Ciencias Aplicadas y Ambientales UDCA and Ministerio de Ciencia, Tecnología e Innovación de Colombia (research project 210484467217).

## Acknowledgements

We thank officials of the Department of Health of Cauca, especially the leader of the Inspection, Surveillance and Health Control Process, Hernando Gil Gómez, for his support in the realization of this research project; to the engineer Anderson Hair Piamba in the ETV Program for his support; and to the vector-borne diseases technicians in the municipality of Patía, Miranda and Piamonte for Ae. aegypti catches. We thank Marco Fidel Suárez for his valuable assistance in performing the fieldwork and his knowledge. We also thank the Genomics of Emerging Microorganisms group of the INS and especially to the Biologist Alicia Rosales Munar for their collaboration with the use of equipment to carry out the processing of the specimens. Lastly, we are deeply grateful to the Sistema de Información sobre Biodiversidad de Colombia (SiB Colombia) for their support in the publishing data process.

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
