## [Editor Report]

Editor’s AssessmentThere’s a shortage of disease vector data available from across the world, and this data release presents a biological collection of Aedes aegypti mosquitoes (the main vector of human arbovirosis) from the department of Cauca, in the southwest of Colombia. In 2022 this group released the first adult mosquito data in GBIF from this region, and here is an update presenting another 160 biological records (3,806 specimens) of the immature stage (pupae) of Ae. Aegypti. Plus some new records (and improved metadata with absence information after feedback). All made publicly available under a CC0 waiver from the Instituto Nacional de Salud and GBIF.

---

## [Reviewer Report]

Upload additional filesDRR-202305-01/form/Data-Review-DRR-202305-01.docxReviewer name and names of any other individual's who aided in reviewer Christopher I HunterDo you understand and agree to our policy of having open and named reviews, and having your review included with the published papers. (If no, please inform the editor that you cannot review this manuscript.)YesIs the language of sufficient quality?YesPlease add additional comments on language quality to clarify if needed
Are all data available and do they match the descriptions in the paper? YesAdditional CommentsMostly, although the dataset includes additional data that is not explicitly described in this manuscript.Are the data and metadata consistent with relevant minimum information or reporting standards? See GigaDB checklists for examples <a href="http://gigadb.org/site/guide" target="_blank">http://gigadb.org/site/guide</a>YesAdditional CommentsBut please note the later comment regarding lack of reporting of negative observations.Is the data acquisition clear, complete and methodologically sound?YesAdditional CommentsIs there sufficient detail in the methods and data-processing steps to allow reproduction?YesAdditional CommentsPlease note the comment below regarding my lack of expertise in the field to know if all the analysis methodology is sufficient.Is there sufficient data validation and statistical analyses of data quality? Not my area of expertiseAdditional CommentsIs the validation suitable for this type of data?YesAdditional CommentsIs there sufficient information for others to reuse this dataset or integrate it with other data?YesAdditional CommentsAny Additional Overall Comments to the AuthorMajor comments (Author action required): 1 – The manuscript describes negative screening events that do no appear to be in the GBIF data? E.g. Table 2 gives the number of containers screened and the percentage +ve with pupae. Without a record of those negative results this summary tables are unreproducible. GBIF has the field “OccurenceStatus” specifically to enable negative observations to be recorded.   Minor comments (Author action suggested): 1 – Event dates are slightly mis-aligned. First event date in dataset is Feb 10th 2021, but the manuscript doesn’t mention any collection dates prior to March 2021. The Feb2021 events are adult occurrences, so it can be seen why there is the difference, but maybe it could be simple enough to add in collection months for the adult data to the manuscript.  2 – Are sufficient methods and data available to reproduce analysis shown in Fig2 and Table3? Please refer to subject area expert reviewers on this matter.  See Data-Review document attached for additional information.
RecommendationMinor Revision

---

## [Reviewer Report]

Upload additional filesDRR-202305-01/form/gx-DR-1683034334 (1).pdfReviewer name and names of any other individual's who aided in reviewer Mauricio dos Santos ConceiçãoDo you understand and agree to our policy of having open and named reviews, and having your review included with the published papers. (If no, please inform the editor that you cannot review this manuscript.)YesIs the language of sufficient quality?YesPlease add additional comments on language quality to clarify if needed
Are all data available and do they match the descriptions in the paper? YesAdditional CommentsAre the data and metadata consistent with relevant minimum information or reporting standards? See GigaDB checklists for examples <a href="http://gigadb.org/site/guide" target="_blank">http://gigadb.org/site/guide</a>YesAdditional CommentsIs the data acquisition clear, complete and methodologically sound?YesAdditional CommentsIs there sufficient detail in the methods and data-processing steps to allow reproduction?YesAdditional CommentsIs there sufficient data validation and statistical analyses of data quality? Not my area of expertiseAdditional CommentsIs the validation suitable for this type of data?YesAdditional CommentsIs there sufficient information for others to reuse this dataset or integrate it with other data?YesAdditional CommentsAny Additional Overall Comments to the AuthorThe text is well written, methodology clear and reproducible. Some observations, in some moments of the text the specimen Aedes aegypti is not in italics, the authors, therefore, should make this small adjustment.RecommendationAccept